# Community Planning Perspective and Its Role within the Social Policy of the Municipalities

**Ivana Butoracova Sindleryova [1,\*], Michal Garaj [1] and Lucia Dancisinova [2]** 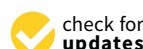

1   University of Ss. Cyril and Methodius, Bucianska 4/A, Trnava 91701, Slovakia; michal.garaj@ucm.sk
2   Faculty of Management, Presov University, Konstantinova 16, Presov 08001, Slovakia;
    lucia.dancisinova@unipo.sk
*   Correspondence: ivana.butoracova@ucm.sk or ivanasindleryova@gmail.com

**Abstract:** Community planning solves policy making at different levels of the state's functioning. Each case has a specific aim and reflects the requirements of citizens. The structure of aims is adjusted to collective and individual levels of clarification of the effect on maximization of potential applications of community planning. The research was realized in the perspective of the municipalities of the Slovak Republic. We used a participatory model of community planning policy as a main tool of community planning potential detection. We emphasized the specific adjustment of community planning to social policy in the Slovak Republic. Data collection was realized by the quantitative research method of a questionnaire. The data were evaluated by a correlation analysis to identify the measure of relationship between variables. Cross-Tabs were used for description of observed categories representation. The results showed a positive effect of the public sector on maximization of community planning policy. On the other hand, the potential declined with rising involvement of the public sector. Individual approach of leading researchers had minimum or no effect on success. It is a collective perspective that decides. In confrontation with the current state, the research represents a unique approach and confirms the exceptionality of the issue. It identifies with the previous findings about a positive contribution of the public participation in community planning. The public perceives the problems critically, expresses the requirements and needs, proposes different solutions and represents important human capital. Additionally, it can make use of contacts and multiply the use of local resources.

**Keywords:** community planning; social policy; public policy; sociology

## 1. Introduction

Public policy provides guidance on how to implement measures in practice, while defining a broad set of theoretical frameworks, philosophical approaches, visions, decisions that transform into concrete programs, projects, or strategies (Khan and Khandaker 2016). The development of theories and their implementation in practice represent the future of the research area development as a lack of theoretical equipment represents a major problem for effective policy performance and requires constant adjustment according to current requirements (Stewart et al. 2008). Local, regional and national politics identify future goals and actions to achieve these goals. It represents the intended set of activities performed by an individual or several actors (Anderson 2010). Implementation of politics and policy requires the cooperation and participation of actors representing different interest sectors. They can be individuals or collective entities that act together and represent relevant actors. The various forms of actors ensure the monitoring of their specific interests. The ways they promote interests are influenced by various factors, with different procedures able to increase or decrease the potential of a particular public policy (May 1993). Sabatier (1988) argues that the public policy maker's argument

lies in the application of instruments and tools that take into account the structure and character of selected target groups of the population for whom the particular public policy is created. The result of the implementation of a strategic plan at the practical level represents the consensus of all involved actors, while at the same time striving for the most advantageous position for a particular subject with regard to the possibilities and requirements of the other actors. Relevant public policy makers at a national level are mainly politicians and civil servants of parliament and government, and may include other representatives of political parties, interest groups, and non-governmental organizations who present solutions to the issues of selected sectors. The success of the effectiveness of public policy implementation highlights the presence of entities that can guarantee expertise and professionalism (Mazmanian and Sabatier 1989). The success and utilization of the potential of the implemented public policy influence in particular the precise handling of the theoretical grasp of the models in practice. Application of standards and elements of selected models allows successful implementation of public policy. Each of the selected models uses different independent variables to influence the overall result. A logical, managerial, organizational, bureaucratic or political model relies on a diverse set of elements to achieve the desired efficiency (Khan and Khandaker 2016). Brinkerhoff and Crosby (2002) argue that success of public policy outputs, resulting strategies and projects depend not only on the effectiveness of the chosen system and managerial skills, but also on the structure and composition of the actors.

*1.1. Community Planning*

Community planning has multidisciplinary character and is used as a tool for strategic planning at local, regional and national levels. The community plan is the main output of the planning process, and is also a binding policy to solve the problem areas. It has a medium-term character and content not only taking into account the needs of cities or municipalities, but being based on the demands of the public. It combines analysis, objectives, implementation in practice, and a control mechanism. Community planning has different settings in different countries. In a foreign context, in Great Britain, Scotland, the United States, or Australia, community-based problems are addressed through community planning, regardless of the area. In these countries, community planning covers a wide complex of sectors such as social policy, health services, education, unemployment, housing policy, family policy, pension policy, youth policy, public transport infrastructure, and development. Community planning policy from the perspective of the Czech and Slovak Republics has an exclusive setting to address social policy and social services (covers also housing, health, education, family policy). Compared to Anglo-Saxon countries, the Czech and Slovak Republics differ in more detailed analytical parts, while the Anglo-Saxon countries pay closer attention to planning efficiency and control mechanisms. In the Slovak Republic, community planning has a specific position when the legislative obligation of all municipal units to process the community plans is applied. Similarly, in the Czech Republic, community planning is regulated by law. Locally, municipal authorities have the right to decide; the principle of voluntariness is applied. At the regional level, similar to the Slovak Republic, the community plan as a strategy also has to be developed in the Czech Republic. At the state level, it is used as the most important strategic planning tool in the social sphere. The state level creates the framework of the planning period for the regions and the local level. It should be stressed that community planning is not regulated within the European Union. On the other hand, sub-tasks and objectives of community planning policy can be addressed through the Structural Funds and European Union programs aimed at these problems. The problem with better understanding of community planning in the Slovak Republic can be caused by different areas of interest of main state bodies. Community planning in the Slovak Republic is coordinated by The Ministry of Labour, Social Affairs and Family. As mentioned above, community planning in the Slovak Republic covers wide complex of areas and policies. Other Slovak ministries that should be included in the process of community planning are not included. We can illustrate it with the Ministry of Health and Ministry of Education. If we look at the regional and local levels in the Slovak Republic, health policy and education policy are also included in community planning. Community planning in the Slovak Republic should be

focused on social policy, but many examples show that health policy, education policy, housing policy, youth policy or family policy are included as well. As a result, community planning in the Slovak Republic is mainly focused on social policy, but with extension to health, education, housing, youth or family policies. Municipalities have the right to regulate and cover all the mentioned areas.

A wide range of participants enter community planning solving. A leading role is played by the representatives of local self-governing bodies, who are the main applicants. Their requirements are the basis of the setting of the entire planning process. When constituting a team of compilers, the contracting authority have to take into account the legislative provisions that talk, at least, about the involvement of representatives of self-governing bodies, recipients of social services, providers of social services and the public. It follows that all three main sectors are involved in community planning policy: public, private and civil. There are also other institutions, organizations or experts in the municipalities and cities that are increasingly involved in strategic planning. The initiative to be involved is either taken by the organizations themselves or the organizations are asked by the research teams. Overall, there is a complex of actors in the community planning policy, as detailed in Table 1.

**Table 1.** Actors involved in community planning policy solving: the case of the Slovak Republic.

| Stakeholder | Actor |
|---|---|
| Self-government | Employees of the city office, deputies (members of political parties, independent deputies), members of the relevant commissions (deputies, experts, the public) |
| Social services provider | Public service provider, non-public social service provider (individuals representing a particular facility, individuals, group representing social service providers) |
| Social services recipient | Social service clients (individual, group representing a particular facility, individual, group representing a group of client equipment) |
| Experts | Methodology of community planning, statistical analysts, expert guarantor |
| Institutions, organizations | Public and private educational institutions, public and private health institutions, non-governmental organizations, civic associations, interest organizations, non-profit organizations from the social sphere, police, schools |
| Public | Individual, a group representing target population groups, population—inhabitants of the city |

Source: Own processing.

The participatory model of community planning (Pilát 2015) represents the most important framework for maximizing utilization of the potential. The model combines several indicators and their presence decides the positive or negative planning policy settings. If the opposite trend occurs, the utilization of the potential is minimized and approaches the administrative model. The differences between the administrative and participatory models of community planning policy are analysed in Table 2. The participatory model studies ways of adopting decisions that are based on the principles of consensus, equality and joint decision-making. It perceives the transparency of the policy solution so that the public has access to all relevant information. At the same time, it allows all actors to have a complete overview. It does not forget about the participation of marginalized groups and recognizes the opinions of all the target groups of the population. It respects primary local resources that are material, financial or human. For understanding, it is important to emphasize that the model primarily takes into account the environment of the Slovak and Czech Republics, but has similar characteristics to Anglo-Saxon countries.

**Table 2.** Participatory vs. administrative model of community planning policy.

| Administrative Model | Participatory Model |
|---|---|
| Hierarchical | Equality of actors |
| Directive | Consensual |
| Unilateral decision-making | Joint decision-making |
| Formalization of certain activities | Real performance of all activities |
| Non-transparent and incomprehensible | Transparent and comprehensible |
| Underestimating of local resources | Maximum utilization of local resources |
| Preference for expert organizations | Real participation of users and the public |
| Limitation or exclusion of users | Effort to involve marginalized groups |

Source: (Pilát 2015), own processing.

Pilát (2015) details and logically argues all indicators of the participatory model. It is a fundamental framework for our research and statistical measurements. In a detailed view of the research, we observe the following indicators of a participatory model: equality of actors/hierarchy; participant consensus/directive approach; joint decision-making/unilateral decision-making; emphasis on the process/emphasis on a paper output; maximizing of the use of local resources/underestimation of local resources; public involvement/non-involvement of the public; involvement of social policy beneficiaries/non-involvement of social policy beneficiaries; involvement of marginalized groups/non-involvement of marginalized groups; regular meetings/irregular meetings; transparency and publication of information/non-publication of information. The first indicator of the pair represents the approximation to the participatory model, while the second is causing the departure towards the administrative model.

The article's main goal is structured into two perspectives. The common feature of both perspectives is to identify how the utilization of the potential of community planning policy is affected. The first view analyzes how the potential of community planning policy can be influenced by different sectors (civic, public and private). The second view is focused on individual perspective and answers the question of how the leader of community planning's policy influenced the maximizing of potential. The first view follows the collective perspective of influence and the second view follows the individual perspective of influence. According to these two views we set two main goals. The first main goal of the article is to identify impact of the sector representation (civil, public, private) on community planning policy to maximize the use of potential. The second main perspective is focused on the clarification of the influence of the leading researcher/manager on maximization of community planning potential. This perspective observes the interconnection of the leading researcher´s/manager's experience from three points of view: experience with community planning, experience with managerial positions, and experience in public administration. The first and second research perspectives seek the linear dependence within the structure of variables. According to main goals we posed a research question: Can the participating sectors or the practical skills of a leading manager influence the fulfilment of participatory model indicators and maximize the potential of community planning policy?

*1.2. Current Research in Community Planning Policy*

The current state of research in the field of community planning policy offers several different approaches. There are articles presenting research results (Santilli et al. 2016; Garnett et al. 2015), analyzing selected cases (Angeles et al. 2014; Pearce 2003; Siemiatycki 2007) or are comparative (Angeles et al. 2014). In addition to these types of studies, we can also find works dealing with methods or applied strategies (Andersson 2011; Garnett et al. 2015). Presentations of research and project results are extended by work manuals and handbooks (Wates 2014) to facilitate the realization of community planning for anyone who starts or wants to improve current practices.

The state of the research from the point of view of quantity of works is not extensive, as it is illustrated by our structured literature overview of studies dealing with the issue. It is necessary to emphasize that the content orientation is not explicitly focused on community planning as a

specific policy that results in a community plan as a strategy. Mostly, community planning is used as a method of involving the public in the decision-making process or of greater public activation. The diversity in approaches to community planning confirms the difference in sectoral perception. It focuses on single selected sectoral policy (Andersson 2011; Santilli et al. 2016; Angeles et al. 2014; Garnett et al. 2015; Siemiatycki 2007; Pearce 2003) or is generalized with the impact on all sector policies (Wates 2014; Pearce 2003). Most often, community planning is used to address health services issues (Andersson 2011; Santilli et al. 2016; Garnett et al. 2015). We also identify cases dealing with social services (Angeles et al. 2014) or transport and infrastructure (Siemiatycki 2007). Multi-sectoral community planning addresses the issues of housing policy, youth policy, infrastructure, health services, social services, unemployment or family policy (Wates 2014; Pearce 2003). Various actors are involved in the process of community planning. Studies do not indicate the number of actors representing specific organizations, institutions, communities and target groups. They rarely use the division of actors into the public, private and civic sectors (Siemiatycki 2007), but rather the studies attempt to highlight outcomes and results achieved. The methodology of actor description is not detailed and included. We did not identify two or more approaches where the involvement of actors was identical. Participation of actors take into account problems and needs of the population. The leading role is given to local parliaments, councils and various officials of the authorities (Andersson 2011; Wates 2014; Garnett et al. 2015; Siemiatycki 2007; Pearce 2003; Angeles et al. 2014). This highlights the position of the public sector. As to the private sector, health services providers and social services providers (Andersson 2011) are those who most often enter the community planning process. The potential involvement of other private actors depends on the area addressed in community planning. The civil sector also plays a prominent role alongside the public sector. It is represented by target groups of populations (Garnett et al. 2015; Santilli et al. 2016), local communities (Wates 2014; Pearce 2003; Angeles et al. 2014; Siemiatycki 2007) or by citizens as individuals. As relevant actors, we also identify service recipients (Andersson 2011), organizations and non-profit organizations (Wates 2014; Santilli et al. 2016; Garnett et al. 2015; Angeles et al. 2014), owners of properties (Pearce 2003) or research institutions (Santilli et al. 2016). A detailed comparative overview of the studies is presented in Table 3.

The fact that we did not find any information about the models that have been applied is an important finding. We obtained the information about a study design, type of community planning, areas of focus, included actors, outcomes and methods. A participatory model of community planning was applied to the research and study. However, we were not able to compare this model to other models. Actually, we can say that the participatory model of community planning is unique. Literature and studies that we reviewed described the community planning in general. The conclusions from literature overview identified positive effects. We identified some which are included also in the participatory model of community planning. Integration of public, transparent partnership, additional resources, reducing of evidence or strength cooperation are the main indicators based on the literature. These indicators represent our interest in the specific Slovak case as well. According to this conclusion we can say that the participatory model is applicable. The resulting indicators are marked as positive every time. That fact confirmed correctness of the participatory model.

**Table 3.** Structured Literature overview summary—community planning in current research.

| Authors | Design of Study | Type of Community Planning/Area | Actors | Outcomes/Methods |
|---|---|---|---|---|
| Andersson (2011) | Article/Research/Methods | Single sectoral/health | Planners at policy level/providers/users in household | Reducing of too much evidence, decisions depend on political interests and political cycles, social audit reduces arbitrariness of planning decisions, social audit training supported by a customized Masters programme |
| Wates (2014) | Handbook/ Manual | Cross-sectoral/ Multi-area | Local governments/local people/ organisations | Additional resources, improve decision-making process, improve access to funding, local people empowerment, strength cooperation, improve education, better insights into communities, improvement of environment shaping |
| Santilli et al. (2016) | Article/Research | Single sectoral/Health | Non-profit organisations/local community/ Research Institution/Aiming groups of peoples | Improvement of community approach method, improvement of the recruitment of participants, enhanced cooperation between stakeholders, increased quality of outputs and recommendations, new visions and projects |
| Garnett et al. (2015) | Article/Research/Methods | Single sectoral/Health | Community coalition/ Non-profit organisations/ Aiming group of people | Increased community engagement, local community empowerment, better understandings of problems, developing of new approaches and strategies |
| Angeles et al. (2014) | Article/ Comaparative analysis/Case Studies | Single sectoral/Social Services | Local government/ municipalities/ local community/ organisations/ providers | Broad comparative perspective, revealing of common features about plan development, implementation, monitoring, reporting and evaluating |
| Pearce (2003) | Article/Case Study | Cross-sectoral/ Multi-area | Local government representatives/ council/property owners/local community | Sustainable hazard mitigation when public is integrated and participate, better problems understanding |
| Siemiatycki (2007) | Article/Case Study | Single sectoral/Rail Infrastructure | Public-private partnership, local community | Improvement of transparent partnership, improvement of better understanding |

Source: Own processing.

## 2. Results

Presentation of the results is structured and reflects the main objectives of the article. The first part of the results evaluates the impact of sectors on maximization of the fulfilment of indicators of community planning potential. It takes into account the active participation of three sectors: public, civil and private. The second part of the results evaluates the action of the main actors or managers of community plans. In this case, we verify the practical skills of team leaders from three perspectives: experience and practice in public administration, management positions and community planning. We combine the multidisciplinary focus of the subject while reflecting the primary areas with which it is directly interconnected.

### 2.1. Results A

Self-governing bodies are the sponsors of community planning in the Slovak Republic, which may result in frequent prevalence of public sector representatives in research teams. The observed case from the perspective of the cities in the Slovak Republic confirmed the assertion by the findings. In 85 cases, public sector representation accounted for more than 50% and in 99 cases it had the highest share compared to other sectors. The dominant position of public sector representatives is enhanced by the presence of cases with more than 90% of the representation (7 cases). The public sector has the largest representation in the share ranges of 40–49.99% (27 cases), 50–59.99% (21 cases) and 60–69.99% (26 cases). The presented results demonstrate the hold of a strong public sector position in strategy decision-making policy in community planning.

The prevalence of the representation of public sector representatives is reflected in the effect of the maximum potential utilization from the community planning. The measurement shows statistically significant results of the relationship between the variables with moderate degree of association (Table 4). The key interpretation of the relationship is presented in Figure 1. It demonstrates that the utilization of community planning potential is increasing with the decline in public sector representation. This finding is an important fact for the process of creation of groups responsible for community planning. The chart identified a case of negative correlation between variables, when the increase in the public sector share negatively influences the potential of community planning. Given presented results about the majority public sector representation and based on the statistical confirmation of the negative correlation, we can claim this finding to be of high importance for future planning periods. The values of the monitored public sector representation cases has a relatively even distribution around the linear line, being primarily located in one main group. There is no division into several groups; an increase or decline in the number of representatives usually mirrors the size of the city by population.

**Table 4.** Correlation—% share of public sector representation and utilization of Community planning of social services potential.

| SUMMARY OUTPUT | | | | | |
|---|---|---|---|---|---|
| *Regression Statistics* | | | | | |
| Multiple R | 0.544953 | | | | |
| Observations | 140 | | | | |
| ANOVA | | | | | |
| | *df* | *SS* | *MS* | *F* | *Significance F* |
| Regression | 1 | 1.344847 | 1.344847 | 58.29435 | $3.38 \times 10^{-12}$ |
| Total | 139 | 4.5285 | | | |

Source: Own processing.

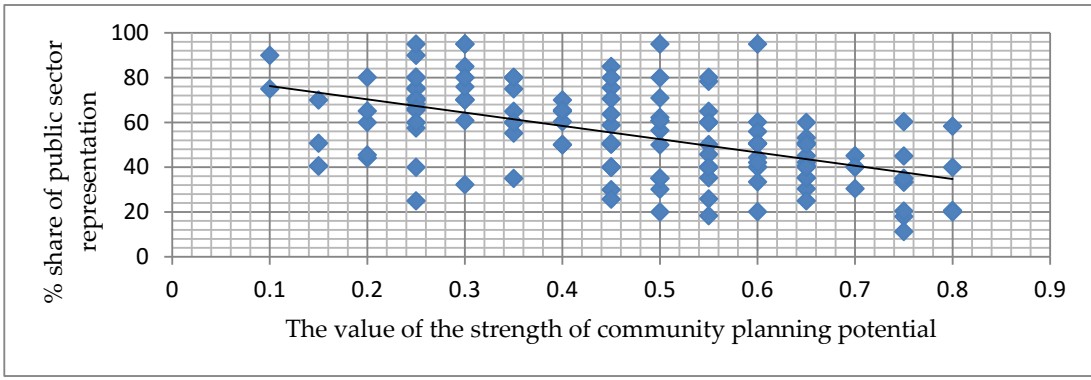

**Figure 1.** Linear line—the case of public sector impact on the utilization of the Community planning potential. Source: Own processing.

The representation of the civic sector in the design and development of the community plan, according to results, proved to be positive in 128 cases, regardless of the size of the collective share. The finding identifies 14 cases where the proportion of the civil sector represented more than 50% of all participants as significant information. The information will be highlighted below with respect to the results of the correlation analysis in the case being measured. The remaining 114 cases showed a representation of less than 50%, and according to a detailed view of the structure, we most often identify 20–29.99% share of the civil sector (30 cases). With an increase and decrease in the proportion of representation, the representation in other perspectives also decreases. The smallest group of cases (9) represent 0.1 to 9.99% of the representation, in other perspectives (10–19.99%; 30–39.99; 40–49.99) we recorded 26 cases on average.

Table 5 evaluates the measuring of the correlation analysis of the existence of a possible impact of the civil sector on the maximization of the result of the implemented community planning. The measuring showed a statistically significant correlation between the variables, with the strength of the association rate moving in the middle range. The above mentioned finding on the percentage of representation of the civil sector highlights a proven statistical measurement of the presence of a positive correlation. It follows from the given result that a team that addresses the issue of community planning of social services should take into account the necessity of civic sector representation to increase the potential of strategic decision-making. According to Figure 2, we can identify a relatively homogeneous group of cases that are distributed near a linear line. Cases are not divided into two or more groups, but are relatively equally represented with respect to decreasing or increasing variables. At the same time, we note that in any of the cases, a problem-solving team did not manage to reach the maximum use of the potential of community planning; on the other hand, there was no opposite, negative case.

**Table 5.** Correlation—% share of civic sector representation and utilization of the Community planning of social services potential.

| SUMMARY OUTPUT | | | | | |
|---|---|---|---|---|---|
| *Regression Statistics* | | | | | |
| Multiple R | 0.59344 | | | | |
| Observations | 140 | | | | |
| ANOVA | | | | | |
| | *df* | *SS* | *MS* | *F* | *Significance F* |
| Regression | 1 | 1.594806 | 1.594806 | 75.01916 | $1.1 \times 10^{-14}$ |
| Total | 139 | 4.5285 | | | |

Source: Own processing.

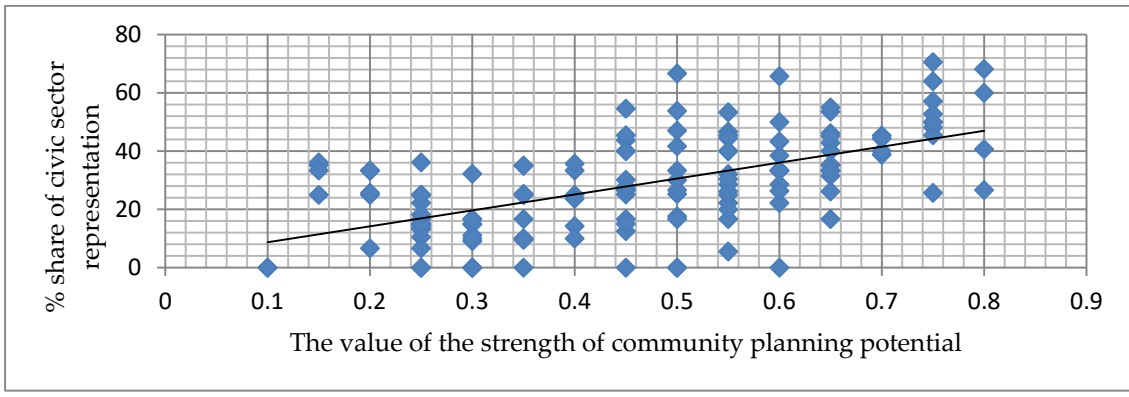

**Figure 2.** Linear line—the case of civic sector impact on the utilization of the Community planning potential. Source: Own processing.

The private sector had considerably lower representation rates than in the first two cases under review. It is caused by the lower number of private facilities providing services in the given city areas, as well as by the fact that one facility is usually represented by one representative or one small group of representatives represents all private facilities. The results show only one case where the private sector achieved the largest share (38.86%). Most often the private sector has only 10–19.99% share of representation (74 cases). Compared to public and civil sectors, the position of the private sector is considerably undersized. This situation is slightly improved by 41 cases with 20–29.99% of representation, but it should be emphasized that only 6 cases in the range of 30–38.86% represent a larger share.

According to Table 6, there is not a statistically significant correlation of the impact of the representation of the private sector on the potential of community planning, whether positive or negative. The association rate is close to the beginning of the range of low strength. We emphasize that this is not a statistically significant correlation. Figure 3 shows an increase in the number of private sector representatives as a positive impact on the use of potential, but in the minimum strength of the association. To identify a positive correlation, the increase in private sector representation could be possible with the decrease in public sector representation in particular. Unlike the public sector, the private sector has a positive, though minimal impact. This statement has only prognostic character based on the direction of the linear line. A successful verification could be possible with a more active participation of the private sector.

**Table 6.** Correlation—% share of private sector representation and utilization of the Community planning of social services potential.

| SUMMARY OUTPUT | | | | | |
|---|---|---|---|---|---|
| *Regression Statistics* | | | | | |
| Multiple R | 0.13497 | | | | |
| Observations | 140 | | | | |
| ANOVA | | | | | |
| | *df* | *SS* | *MS* | *F* | *Significance F* |
| Regression | 1 | 0.082495 | 0.082495 | 2.560569 | 0.111846 |
| Total | 139 | 4.5285 | | | |

Source: Own processing.

The common perspective of correlations from measured cases is presented in Figure 4. It defines a summary of the direction of impact on community planning potential. We observed the existence of two positive correlations in the low and medium strengths of the association. We draw attention to

the statistical significance of the positive impact on maximizing the potential of community planning only from a civil sector perspective. The second option is represented by the impact of public sector representation, but in a negative sense. In this case, on the other hand, we are talking about reducing the use of community planning potential while increasing the share of public sector representation. The third case of the private sector is a separate category because of the statistical insignificance. Primarily in terms of the smallest representation, it is also the lowest level of association of the impact of community planning potential. As we mentioned, more objective results in the case of the private sector would be gained by its greater involvement. With regard to civic and public representation, the private sector does not have a balanced position.

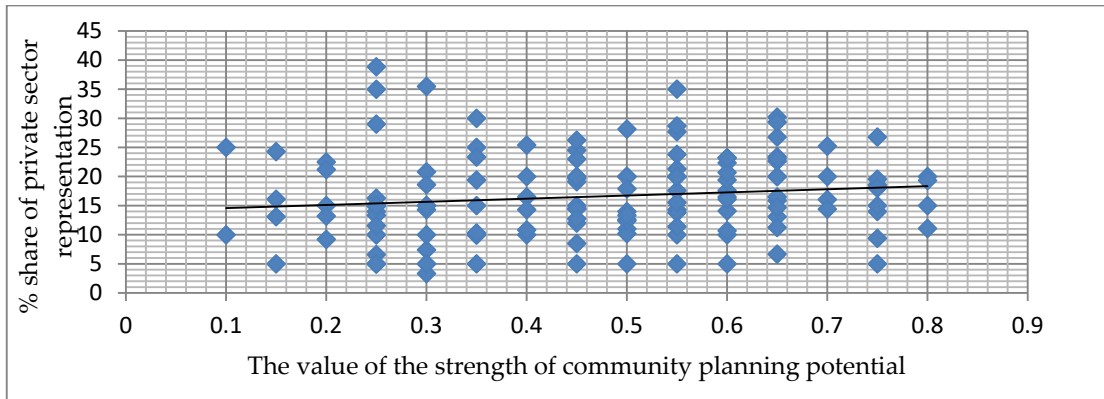

**Figure 3.** Linear line—the case of private sector impact on the utilization of the Community planning potential. Source: Own processing.

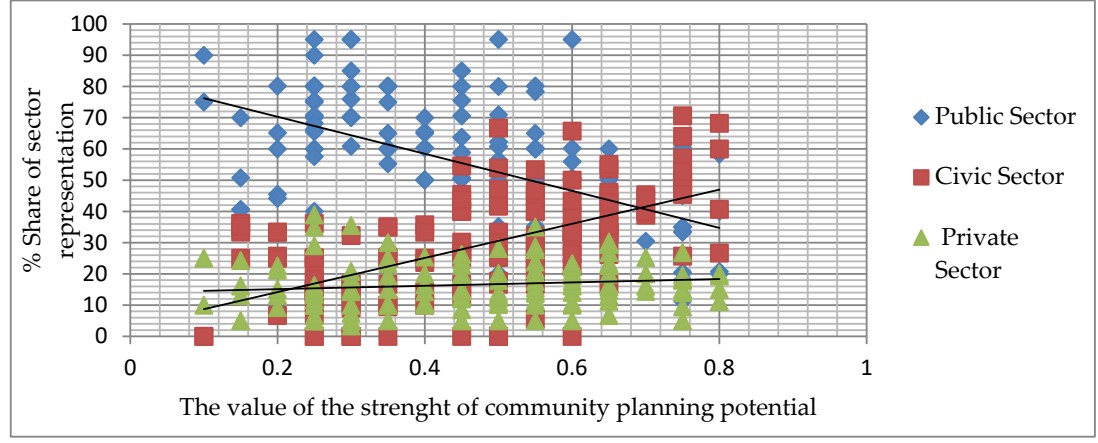

**Figure 4.** Linear line—comparison of the impact of the sectors on the KPSS potential. Source: Own processing.

*2.2. Results B*

The second part of the results is focused on the impact of selected senior managers'/leading researchers' experience on the possible maximization of community planning potential. While the first part of the results studied the impact of the team on the measured variable, the second part of the results seeks to clarify the potential impact of an individual on maximizing the potential of community planning. We evaluate acquired skills in terms of a number of years. As community planning is a multidisciplinary issue, we observe the experience from each of them: public administration, community planning and management. As in the first case of sector representation, in addition to showing the distribution of individual cases, we observe the statistical significance of correlations.

In the first measurement (Table 7), we evaluate the experience with community planning of social services. Although community planning has been legally supported since 2008, the smallest group is

represented by respondents with 10 or more years of experience. It can be caused by several reasons: frequent personnel changes in the teams of actors, failure to comply with the legislative obligation to develop a community plan, staff exchanges in local parliaments, and the composition of commissions following local elections. Local councils´ commissions are most often responsible for the creation of a community plan, resulting in a frequent dominance of the public sector in addressing the issue. Regardless of the number of years of experience with community planning from previous periods, the most frequent representation of the impact of the potential is in the middle range. A similarly important factor is the decrease in the incidence of cases of increasing number of years of practice. Despite the fact that the highest percentage of maximum utilization of community planning is found in respondents with 10 or more years of experience, this finding is limited by the low number of cases. Compared to the perspective of practice in public administration and managerial positions, the opportunity to gain experience with community planning is very limited. We stress again that the legislative obligation to create community plans in the Slovak Republic has existed only from 2008.

**Table 7.** Experiences with Community planning × Community Planning Potential (count, row %,column %, total %).

| Experiences with Community Planning | Community Planning Potential | | | | Total |
|---|---|---|---|---|---|
| | 0–0.1: Very Low | 0.11–0.33: Low | 0.34–0.66: Middle | 0.67–1: High | |
| 0–5 years | 1<br>1.35%<br>33.33% | 25.00<br>33.78%<br>67.57% | 42<br>56.76%<br>50.60% | 6<br>8.11%<br>35.29% | 74<br>100.00%<br>52.86% |
| 6–9 years | 1<br>2.08%<br>33.33% | 11<br>22.92%<br>29.73% | 31<br>64.58%<br>37.35% | 5<br>10.42%<br>29.41% | 48<br>100.00%<br>34.29% |
| 10 and more years | 1<br>5.56%<br>33.33%<br>0.71% | 1<br>5.56%<br>2.70%<br>0.71% | 10<br>55.56%<br>12.05%<br>7.14% | 6<br>33.33%<br>35.29%<br>4.29% | 18<br>100.00%<br>12.86%<br>12.86% |
| Total | 3<br>2.14%<br>100.00%<br>2.14% | 37<br>26.43%<br>100.00%<br>26.43% | 83<br>59.29%<br>100.00%<br>59.29% | 17<br>12.14%<br>100.00%<br>12.14% | 140<br>100.00%<br>100.00%<br>100.00% |

Source: Own processing.

Measuring of the correlation of impact did not show statistically significant values between variables. The results are presented in Table 8 according to which we identify the minimum, or almost no relationship between community planning experience and maximizing community planning potential. Previous experience of the chief manager, or researcher does not directly influence the positive results of community planning.

**Table 8.** Model Summary (Community Planning Potential), ANOVA (Community Planning Potential).

| R | R Square | Adjusted R Square | Std. Error of the Estimate | | |
|---|---|---|---|---|---|
| 0.21 | 0.04 | 0.04 | 0.65 | | |
| | Sum of Squares | df | Mean Square | F | Sig. |
| Regression | 2.66 | 1 | 2.66 | 6.27 | 0.013 |
| Residual | 58.51 | 138 | 0.42 | | |
| Total | 61.17 | 139 | | | |

Source: Own processing.

In the second measurement (Table 9), we examine the experience of executive managers with management positions. Compared to the case of community planning experience, the quantitative expression of years of practice is vastly different. It is mainly due to the possibility of acquiring managerial positions in different social spheres, while community planning has limited possibilities. We evaluate managerial practice in 10-year intervals, with a relatively equal representation in the ranges of 0 to 9 and 10 to 19 years. As in the case of community planning, we record the highest percentage representation in the middle range of the utilization of community planning potential. We see positive representation of respondents with 20 or more years of managerial experience, we have zero representation in a very low range and minimal representation in a low range of community planning potential utilization.

**Table 9.** Experiences as Manager × Community Planning Potential (count, row %, column %, total %).

| Experiences as Manager | Community Planning Potential | | | | Total |
| --- | --- | --- | --- | --- | --- |
| | 0–0.1: Very Low | 0.11–0.33: Low | 0.34–0.66: Middle | 0.67–1: High | |
| 0–9 years | 2 <br> 2.94% <br> 66.67% | 17 <br> 25.00% <br> 45.95% | 43.00 <br> 63.24% <br> 51.81% | 6 <br> 8.82% <br> 35.29% | 68 <br> 100.00% <br> 48.57% |
| 10–19 years | 1 <br> 2.00% <br> 33.33% | 17 <br> 34.00% <br> 45.95% | 26 <br> 52.00% <br> 31.33% | 6 <br> 12.00% <br> 35.29% | 50 <br> 100.00% <br> 35.71% |
| 20 and more years | 0 <br> 0.00% <br> 0.00% | 3 <br> 13.64% <br> 8.11% | 14 <br> 63.64% <br> 16.87% | 5 <br> 22.73% <br> 29.41% | 22 <br> 100.00% <br> 15.71% |
| Total | 3 <br> 2.14% <br> 100.00% | 37 <br> 26.43% <br> 100.00% | 83 <br> 59.29% <br> 100.00% | 17 <br> 12.14% <br> 100.00% | 140 <br> 100.00% <br> 100.00% |

Source: Own processing.

Statistical verification of the possible presence of correlation between variables follows the previous measured case. Similarly, according to Table 10, managerial practice of respondents does not show statistically significant values. The strength of the association is located in a very low range or the resulting value is close to 0. The case when the level of managerial practice of a leading researcher has no influence on the success of the maximum utilization of the potential has been reaffirmed. Despite wider opportunities to achieve managerial positions, these do not play an important role in fulfilling indicators of a participatory community planning model. We return back to the argument that the quantitative and qualitative composition of the team of actors is the decisive factor.

**Table 10.** Model Summary (Community Planning Potential), ANOVA (Community planning Potential).

| R | R Square | Adjusted R Square | Std. Error of the Estimate | | |
| --- | --- | --- | --- | --- | --- |
| 0.12 | 0.02 | 0.01 | 0.66 | | |
| | Sum of Squares | df | Mean Square | F | Sig. |
| Regression | 0.96 | 1 | 0.96 | 2.19 | 0.14 |
| Residual | 60.22 | 138 | 0.44 | | |
| Total | 61.17 | 139 | | | |

Source: Own processing.

The last measurement focuses on the experience in the public administration or local government. It is a category where we record cases of respondents with the highest numbers of years of practice.

This fact is reflected in the resulting data set with four ranges of years of practice. Some of the respondents started in the field of public administration before the social-political changes in the Slovak Republic in 1989. In this context, it has an exceptional position and the lowest limitations compared to other cases. Respondents (Table 11), as in the previous two cases, are located in the middle range of potential utilization regardless of their years of practice. A positive finding is the minimal occurrence of very low values of potential utilization. Overall, according to the structure of the data set, each range is well balanced.

**Table 11.** Experiences in public administration × Community planning Potential (count, row %, column %, total %).

| Experiences in Public Administration | KPSS Potential | | | | Total |
|---|---|---|---|---|---|
| | 0–0.1: Very Low | 0.11–0.33: Low | 0.34–0.66: Middle | 0.67–1: High | |
| 0–9 years | 1 2.50% 33.33% | 10 25.00% 27.03% | 25 62.50% 30.12% | 4 10.00% 23.53% | 40 100.00% 28.57% |
| 10–19 years | 1 1.89% 33.33% | 20 37.74% 54.05% | 25 47.17% 30.12% | 7 13.21% 41.18% | 53 100.00% 37.86% |
| 20–29 years | 1 2.63% 33.33% | 5 13.16% 13.51% | 30 78.95% 36.14% | 2 5.26% 11.76% | 38 100.00% 27.14% |
| 30 or more years | 0.00% 0.00% | 2 22.20% 5.41% | 3 33.30% 3.61% | 4 44.40% 23.53% | 9 100.00% 6.43% |
| Total | 3 2.14% 100.00% | 37 26.43% 100.00% | 83 59.29% 100.00% | 17 12.14% 100.00% | 140 100.00% 100.00% |

Source: Own processing.

The clarified trend in the previous two measurements indicated the likely development of the last measurement. The results of the correlation analysis confirmed the previous two cases and also did not show statistically significant findings. The impact of practice of a leading researcher in public administration (Table 12) is even more insignificant than in the case of managerial positions. The observed case is consistent with the finding that the practice in public administration of an individual is not able to positively influence the potential of fulfilling the indicators of a participatory community planning model. On the other hand, it should be pointed out that the findings do not show any negative impact either; the perspective of the respondents' skills acquires a neutral position without significant positive and negative impact.

**Table 12.** Model Summary (Community planning Potential), ANOVA (Community planning Potential).

| R | R Square | Adjusted R Square | Std. Error of the Estimate | | |
|---|---|---|---|---|---|
| 0.12 | 0.01 | 0.01 | 0.66 | | |
| | **Sum of Squares** | **Df** | **Mean Square** | **F** | **Sig.** |
| Regression | 0.89 | 1 | 0.89 | 2.03 | 0.15 |
| Residual | 60.29 | 138 | 0.44 | | |
| Total | 61.17 | 139 | | | |

Source: Own processing.

## 3. Discussion

The results of the research demonstrated two different perspectives of the importance of participating in community planning. With various levels of success, a team or an individual

contribute to the final processing of the strategy. While we have identified partial positive impacts for the team, there is a strong, one-way negative trend for individuals. The team on behalf of the civil sector is the only perspective that has the potential for success in the future. This is evidenced not only by the numerous analyzed studies, but also by the results of the statistical measurements carried out. It should be noted that in the case of the Slovak Republic we are talking about the obligation to involve the public in the process of community planning policy. While the other analyses do not have this obligation, nevertheless, their results emphasize the need to involve the public as much as possible. The positive impact of the civil sector is in absolute contradiction with the public sector. It is not possible to deny the elected representatives and employees of the state authorities the work effort, but ultimately, they cannot fully reflect the real needs of the population. Measurement results confirm that increasing public sector participation tends to enforce non-consensual and directive decisions. Ultimately, it minimizes the effectiveness and efficiency of community planning policy. The private sector's perspective is characterized by the peculiarity of limited possibilities. We are not only talking about the lower involvement of representatives, but also about the fact of ignoring the possibility to participate from the public sector perspective. Executive managers as individuals do not have a significant share in maximizing the utilization of policy potential. Their position is limited not only by the presence of a wider team, but mainly by the decisions and demands of the contracting authority. Moreover, representatives of the civil sector are also actively involved in the decision-making process. The leaders of the community planning process teams are thus primarily responsible for managing and organizing a group of people.

Future developments in community planning policy should reflect the current state of research. The positive impact of the public and citizens in the process of developing the strategy is unambiguously confirmed. They represent not only a creative element, but are also active critics. The public sector participation should be reduced to increase the activity of the private sector. However, we must keep in mind that community planning is a specific issue. The same way as it is possible to create a unique setting for a selected area, it is also possible to have a unique composition of a team of actors. Regardless of the current state of research and the results of statistical measurements, community planning policy represents a future in taking into account the demands of the population while taking effective measures.

The strength of study lies in the uniqueness of the participatory model that we used. Actually, it is not possible to identify a similar study from the available literature and research. The study uniqueness is highlighted by the used methods and applied model approach. The complexity of community planning creates the possibility to be applied in other planning areas. We are talking about the broadest possible involvement of the civil sector in the strategic planning process. Ideally, the leader of a team should already consider the involvement of the civic sector. Overall according to study results, our approach is applicable to any strategic planning. But the primary focus should be on local governments, municipalities or regional governments. The research results opened the opportunity for the future continuation to see possible changes. The research can continue from city to municipality level. But it will be much more difficult to execute research at the municipality level. On the other hand, in case of the Slovak Republic where there are more than 2900 municipalities, the results will be more representative and detailed. Cross-national study could be processed in comparison with the Czech Republic because of the similar history and development of community planning. Other comparisons could be realized from the point of view of used methods or approaches. The weaknesses of the study may be reflected in several perspectives. From the view of data collection, a questionnaire was always filled in by the manager. That means he or she could not be challenged by his or her colleagues when answering the questions. We also admit the absence of closer personal contact with respondents. It is also possible to criticize the used participatory model of community planning from the point of view of indicators, which has not yet undergone more fundamental criticism of other authors. As a limitation we can also mention that the study is a specific case. On the other hand, we stress again the uniqueness of the used approach.

In comparison with results of the structured literature overview, we found some similarities and differences in our research. As the main conclusion we stress the deficit of models which are used in community planning. The participatory model of community planning is actually the only one. Some indicators of participatory models are also included among positive effects from literature overview. Our case analyzed the presence of all indicators of participatory model of community planning. The added value of this study is that we showed positive impacts on a potential of community planning when a civic and private sector are included in the research teams. The positive impact of civic sector on a community planning confirms the findings of several studies (Wates 2014; Garnett et al. 2015; Pearce 2003). Each of 140 cases included in the research represents a specific case. Structured literature overview showed the same fact in Santilli et al. (2016); Garnett et al. (2015); Angeles et al. (2014); Siemiatycki (2007) and Andersson (2011). There is not an equal approach, method or model used in community planning. Conclusions present positive effects of community planning. This study also deals with an extra issue of how the potential of community planning policy can be affected by the inclusion of the public, civic or private sectors as well as how a leading manager´s experience can influence the potential of community planning policy. These two new research approaches are unique and take the participatory model of community planning to the next level.

## 4. Materials and Methods

Methodology of the article processing and solving tasks in relation to its objectives uses separate research strategies to work with both secondary sources and primary data. The secondary sources of literature that are contained in the introductory part are processed through the use of the Structured Literature Overview Strategy. Searching for the secondary sources of literature dealing with current state of the issue is done through a combination of keywords. The literature search process is carried out in scientific databases and other available resources. The limitation is a search in freely accessible databases, websites and paid databases to which the workplace has access. Key words for searching in databases are presented in Table 13; we gain a wider space for identifying appropriate studies to address issues by their mutual combination.

**Table 13.** Key words for identifying key literature of current research.

| Participant | Measurement | Outcomes | Study Design |
|:---:|:---:|:---:|:---:|
| actors | planning | Engaging | |
| sectors | participation | Practice involving | Comparative |
| public | impact | decision-making process | Methodological |
| private | effect | developing | Case studies |
| civic | influence | social services plan | Research |
| community | forcing | model of participation | |

Source: Own processing.

In the initial search phase, we identified more than 500 possible published sources in diverse forms such as articles, books, strategy papers, and legislation. For the purposes of the research situation solving, the number of the identified content was reduced to 7 key sources, which we discussed in the introductory part of the article. The following part of the article presents the findings of the realized research, while the results directly reflect the specified main and partial objectives/tasks. To solve the tasks, we use a quantitative research strategy using standardized questionnaire techniques. We used this strategy because research covers all Slovak cities and we are able to collect the data from each of them. The basic sample of respondents is determined in a deliberate manner, when the implementation takes place in all self-governing units in the territory with the status of a city.[1] The selection sample

---

[1] Till 5 April 2019, there exist 140 self-government units with the statute of a city at the territory of the Slovak Republic (Statistical Office of Slovak Republic 2019).

of respondents copies the basic sample of respondents. A respondent from each of the 140 cities in the Slovak Republic is represented by the leading researcher/manager of social services community planning. It is important in this case to point out that the process of creating community planning takes place with the participation of many actors representing different sectors. The leading researcher or manager represents for us the most relevant representative who is addressed as a respondent. The structure of the standardized questionnaire has included four sections, where the first one deals with sociological and demographic data about respondents; the second identifies the representation of actors and sectors in the community planning process; the third identifies indicators of participatory model of community planning potential and the last one deals with the experience of the leading researcher/manager in the field of community planning, self-government and managerial positions. The indicators of participative model of community planning we discussed in the theoretical part of study. Contact with respondents was in the form of a personal meeting, telephone communication or via electronic (email). 100% of questionnaires were returned answered and completed by the respondents. The entire process of data collection was carried out in several phases until 100% response was obtained from all respondents, as the respondents were deliberately set and no substitution was possible.

The processing of respondent data is done through the Microsoft Excel statistical software, where we create the basic data matrix of all responses, while the selected responses that have a qualitative character are transformed into quantitative form. We use correlation analysis to verify the existence of a relationship between variables and the measure of association. We apply the correlation analysis in the case of the relationship between sector participation and the potential of community planning of social services analysis; for the graphic illustration we also present the linear regression curve. The impact of the leading researcher's/manager's experience on the potential of community planning is addressed through the Cross-Tabs T-Test with Pearson's correlation coefficient confirmation. Both cases are measured in the SPSS statistical software, in which we use a processed data matrix from Microsoft Excel software.

The calculation of community planning potential maximization is based on a participatory community planning model that is discussed in detail in the introductory section. We identify individual indicators of participatory model through questionnaires and their presence, or absence affects the resulting value. The formula for calculating the value of community planning potential is as follows:

$$\frac{(number\ of\ identified\ indicators)}{(total\ nubmber\ of\ inticators - n)} = \text{value of the strength of community planning potential}$$

The maximum presence of all indicators demonstrates the full use of the participatory model of community planning and the potential of community planning of social services. The strength of community planning potential takes into account four levels: none/minimum (0–0.1) small (0.11–0.33), medium (0.34–0.70), high (0.71–1). The strength value of the potential of community planning of social services is an important indicator for subsequent measurements following sub-objectives. The following calculations of the statistical significance of the existence of the relationship between variables and association rates between them again take into account four strength levels: none/minimum (0–0.1) small (0.11–0.33), medium (0.34–0.70), high (0.71–100). Since we test for the presence of a relationship between variables based on the principle of randomness without previous assumptions, we therefore used the correlation analysis using Pearson's correlation coefficient. We measure the strength of association between the community planning potential and two other variables: the representation of the sectors in the research team and the leading researcher's experience. In the case of the sector representation variable, we perform four measurements that analyzed the overall number of representations of all sectors, the share of public sector representatives, the share of private sector representatives and the share of representatives of the civil sector. An important aspect in the case of sector representation is the calculation based on the percentage of sector representation from the total number of all representatives in the research team. In the case of the leading researcher's/manager's

experience, we carry out three measurements that examine: the number of years of experience in public administration, the number of years of community planning experience, and the number of years of experience with management positions. The second variant of the measurement of leading researcher's/manager's experience is realized through use of the Cross-Tabs T-Test. We divide the individual specific kinds of experience into several categories according to the number of years.

**Author Contributions:** Author contributions statements as follow: conceptualization: I.B.S., M.G. and L.D.; methodology: M.G.; I.B.S.; software: M.G.; validation: L.D. and I.B.S.; formal analysis: M.G.; data curation: M.G.; writing—original draft preparation: M.G.; resources: M.G.; writing—review and editing: L.D. and I.B.S.; supervision: L.D. and I.B.S.; visualization: L.D. and I.B.S.

**Funding:** This research received no external funding.

**Conflicts of Interest:** The authors declare no conflict of interest.

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
