# Peer review of "Community Planning Perspective and Its Role within the Social Policy of the Municipalities"

_socsci, doi:10.3390/socsci8060183_

Round 1
Reviewer 1 Report
Thank you for the opportunity to review the manuscript “Community planning perspective and its role within the social policy of the municipalities” for Social Sciences. This is a thought-provoking manuscript, and I enjoyed reviewing it. The data are unique and allow the authors to examine (in an exploratory fashion) “lessons learned” from research regarding different perspectives on the importance of participating in community planning. Overall, I found the manuscript to be thought provoking. I do, however, have some comments on how the manuscript could be improved.
1. The paper is in need of basic editing, as its present form makes for a very distressing read and at times interferes mightily with comprehension.
2. The literature is appropriately reviewed and establishes a meaningful theoretical context for the research
3. The tables and charts needs to be formatted for easier comprehension. For example, the inclusion of the numerical information in the charts is, substantively, meaningless with regards to the visualization. In fact, I believe that the numbers create the illusion of a clustering effect that is likely not actually present within these data.
4. There is no discussion of the strengths and weaknesses of this study. This seems to be an easily fixable but glaring omission on the part of the authors.
In all, this was an interesting paper and has the potential to make a nice contribution to the literature; I look forward to seeing a revised version
Author Response
Response to Reviewer 1 Comments
Point 1: The paper is in need of basic editing, as its present form makes for a very distressing read and at times interferes mightily with comprehension.
Response 1: The entire text of the article has been newly edited. The sentences have been reformulated and simplified. The English language of the article (both, a form and style) was proofread by a native speaker and the mistakes in the interpretation of the authors were removed. The whole text was edited to make the study more understandable and distressing for readers.
Point 2: The literature is appropriately reviewed and establishes a meaningful theoretical context for the research.
Response 2: The literature – no changes necessary according to the reviewer opinion.
Point 3: The tables and charts needs to be formatted for easier comprehension. For example, the inclusion of the numerical information in the charts is, substantively, meaningless with regards to the visualization. In fact, I believe that the numbers create the illusion of a clustering effect that is likely not actually present within these data.
Response 3: The charts and tables were edited and formatted for easier comprehension. The distressing visualization was unfortunately caused by the formatting of the text in the final submission, we feel terribly sorry for that. Columns have been modified and merged in Table 3, reducing the table range. The content was retained. Tables: 4, 5, 6 were formatted to a smaller version. Tables 7, 9, and 11: removed unnecessary data and are also formatted into a smaller version. Tables 8, 10 and 12 were formatted into a smaller version. Charts: 1, 2, 3, 4 were replaced by new ones. The format of the original charts was changed by uploading system, which the authors could not influence.
Point 4: There is no discussion of the strengths and weaknesses of this study. This seems to be an easily fixable but glaring omission on the part of the authors.
Response 4: The strengths and weaknesses of the study were added to the text in the Discussion section.

Reviewer 2 Report
The paper is about a highly topical theme and it potentially is of great interest for the journal. However it has many limits. On the whole, several important concepts seem take for granted, so it is difficult to understand research data and discussion.
Particularly:
- It is appropriate to define main goals more clearly and it is necessary to specify methodological choices. What are the topic of the survey and the indicators to detect them? What do exactly describe terms of the formula (par. 4)? The numerator is the “number of identified indicators” o the “number of unidentified indicators”? How has been collected the ones and the others? As denominator what is “n”? A more detailed explanation helps a better discussion on results and a their better comprehension.
- p. 2: in different countries the same concepts have different contents, so it is necessary to specify them. The comparative analysis among countries seems somewhat confused, it should be more clear and referring to similar issues for each analyzed context. For example, why “social policy” is separated from “housing policy”, “family policy”, etc.? What are its contents, different from the other policies?
- On the “community planning policy” certain assumptions seem taken for granted, instead they are useful for understanding the contents. For example, the work analyses the presence of the three main sectors (public, private and civil) in community social services. But community planning can take many forms and involved actors can take several roles (co-planner, consultant, facilitator, executor,…), which are never specified.
- In Table 2: “formalization of certain activities” seems against “transparent and comprehensible”. Why? They are not antithetical concepts
- Table 3 shows the different approaches used in research on the community planning policies, but the several findings are not debated later. What is its purpose? It could be aimed at define models of community planning policy, to compare with research results.
- The par. 4 “Materials and Methods” is the methodological description of the search, so it should precede its results, to facilitate their comprehension.
- Tables and Charts should give an easy reading of data. Charts are too confusing, unreadable, all numbers should be deleted to maintain only the most representative. Tables 7, 9 and 11 show too many percentage. They have only to show that ones useful for an easy comprehension of results.
Author Response
Point 1: It is appropriate to define main goals more clearly and it is necessary to specify methodological choices. What are the topic of the survey and the indicators to detect them? What do exactly describe terms of the formula (par. 4)? The numerator is the “number of identified indicators” o the “number of unidentified indicators”? How has been collected the ones and the others? As denominator what is “n”? A more detailed explanation helps a better discussion on results and a their better comprehension.
Response 1: The main objectives were modified and reformulated, the description of the questionnaire was completed and modified, the indicators of the participatory model are explained in Table 3 in the Community planning. The formulae for calculating the value of the potential of community planning was modified, the study only monitored the identified indicators of a participatory community planning model.
Point 2: in different countries the same concepts have different contents, so it is necessary to specify them. The comparative analysis among countries seems somewhat confused, it should be more clear and referring to similar issues for each analyzed context. For example, why “social policy” is separated from “housing policy”, “family policy”, etc.? What are its contents, different from the other policies?
Response 2: In community planning, the members represent different institutions and sectors. Subsequently, they have different positions and perform different roles, but the aim of the study was not to examine their roles and positions. Each processing team creates its own organizational structure that was not focused on in our research. The study is interested in the composition of a team of builders in terms of sector representation, does not address the roles and functions of members.
Point 3: On the “community planning policy” certain assumptions seem taken for granted, instead they are useful for understanding the contents. For example, the work analyses the presence of the three main sectors (public, private and civil) in community social services. But community planning can take many forms and involved actors can take several roles (co-planner, consultant, facilitator, executor,…), which are never specified.
Response 3: The authors completed the explanation of a community planning in the Slovak Republic and abroad, an explanation of why some other policies are included in the community planning in the Slovak Republic was added to the article, we also present an explanation of the difference between the highest state level (Ministries) and municipalities.
Point 4: In Table 2: “formalization of certain activities” seems against “transparent and comprehensible”. Why? They are not antithetical concepts
Response 4: The table has been modified. There was a mistake in the first version when entering the indicators.
Point 5: Table 3 shows the different approaches used in research on the community planning policies, but the several findings are not debated later. What is its purpose? It could be aimed at define models of community planning policy, to compare with research results.
Response 5: The sense of approaches from the literature and research points to their diversity, the literature review is also focused on the use of models, but no other model approach was reported in any of the sources, this fact and the findings have been added to the study. The discussion was extended to confront our study results with the findings of Structured Literature Overview.
Point 6: The par. 4 “Materials and Methods” is the methodological description of the search, so it should precede its results, to facilitate their comprehension.
Response 6: Position of Materials and Methods are set in the template. We only follow the study template requirements.
Point 7: Tables and Charts should give an easy reading of data. Charts are too confusing, unreadable, all numbers should be deleted to maintain only the most representative. Tables 7, 9 and 11 show too many percentage. They have only to show that ones useful for an easy comprehension of results.
Response 7: The charts and tables were edited and formatted for easier comprehension. The distressing visualization was unfortunately caused by the formatting of the text in the final submission, we feel terribly sorry for that. Columns have been modified and merged in Table 3, reducing the table range. The content was retained. Tables: 4, 5, 6 were formatted to a smaller version. Tables 7, 9, and 11: removed unnecessary data and are also formatted into a smaller version. Tables 8, 10 and 12 were formatted into a smaller version. Charts: 1, 2, 3, 4 were replaced by new ones. The format of the original charts was changed by uploading system, which the authors could not influence.

Round 2
Reviewer 2 Report
Contents are now more detailed and the unclear questions are specified and argumented.